# Bilateral Optic Neuritis after COVID-19 Vaccination: A Case Report

**DOI:** 10.3390/vaccines10111889

**Published:** 2022-11-09

**Authors:** Ching-Chih Liu, Wan-Ju Annabelle Lee

**Affiliations:** 1Department of Ophthalmology, Chi Mei Medical Center, Tainan 71004, Taiwan; 2Graduate Institute of Medicine, College of Medicine, Kaohsiung Medical University, Kaohsiung 80708, Taiwan; 3School of Pharmacy, Institute of Clinical Pharmacy and Pharmaceutical Sciences, College of Medicine, National Cheng Kung University, Tainan 70101, Taiwan

**Keywords:** COVID-19, vaccine, AZ vaccine, ChAdOx1, optic neuritis

## Abstract

Background: Neuro-ophthalmic manifestations after vaccines are rare, with optic neuritis (ON) being the most common presentation. Patients with vaccine-related ON are similar to those with idiopathic ON. The temporal relationship between vaccination against and the occurrence of ON is vital. Here, we report a case of bilateral ON after the administration of the ChAdOx1-S nCoV-19 SARS-CoV-2 vaccine. Case: A 49-year-old healthy Asian female presented with sudden onset of bilateral blurred vision within 2 days. She complained of photophobia and extraocular pain upon movement over 3 days. Upon examination, her best corrected visual acuity (BCVA) was 20/30 in the right eye and 20/200 in the left eye. Anterior segment findings were unremarkable, with normal intraocular pressure. Fundoscopic examination revealed bilateral disc edema with vessel engorgement. Visual field examination revealed profound visual field defect in both eyes. She denied any trauma, use of new medication or medical history. She had received the ChAdOx1 nCoV-19 SARS-CoV-2 vaccine 14 days prior. Under suspicion of vaccine-related optic neuritis, she was given intravenous methylprednisolone 1 gm/day for 3 days, shifting to oral prednisolone under gradual tapering for 2 weeks. Conclusions: Typically presenting with sudden-onset visual decline and extraocular pain during movement, acute ON is generally idiopathic. Bilateral ON is rare, but quick identification is important because it can potentially lead to permanent loss of vision if left untreated. Vaccination-induced ON is even rarer but not difficult to treat. However, such patients require further evaluation and long-term follow-up because they may be prone to other neurological disorders in the future.

## 1. Introduction

Since emerging in November 2019, COVID-19, or Coronavirus Disease 19, has become a priority healthcare issue. Induced by the SARS-CoV-2 virus, it presents a diverse variety of mainly respiratory symptoms, which can range from the asymptomatic to serious, life-threatening conditions, collectively referred to as severe acute respiratory syndrome. To date, it has claimed over 6 million deaths worldwide. As the COVID-19 pandemic continues, the development of vaccines against this virus has been crucial to prevent morbidity and mortality. To date, more than six vaccines have been approved for COVID-19 treatment (BNT162b2, mRNA-1273, ChAdOx1-S, BBV152 (Covaxin), Ad26.COV2.S, and Sputnik V (Gam-COVID-Vac)). More than 12.5 billion people worldwide have received these vaccines.

Optic neuritis, an inflammatory optic nerve disorder, is relatively infrequent with a prevalence of only 0.1%. Its primary clinical characteristic is the rapid (1–2 days) onset of loss of vision, and it is aggravated by heat or exercise and is frequently accompanied by eye pain and/or headache [1,2]. Common symptoms and signs include reduced visual acuity, ocular pain during eye movement, reduced brightness perception, and altered color vision. While not common, the development of post-vaccination optic neuritis has occasionally been reported [3,4,5,6,7,8,9].

Since the widespread administration of vaccines against SARS-CoV-2, several potential adverse effects and diseases caused by the vaccine have been reported. In March 2021, there were several reports of patients experiencing thrombosis events with thrombocytopenia after AZ vaccination [10]. Cases of neurological or neuro-ophthalmological manifestations after COVID-19 vaccinations were also observed and have been published. One case of neuromyelitis optical spectrum disorder (NMOSD) after coronavirus mRNA-1273 vaccination has been reported [11]. Another case involving simultaneous optic neuritis and thyroiditis after SARS-CoV-2 vaccination has been reported [12]. A few adverse neuro-ophthalmic case report series related to COVID-19 vaccines have also recently emerged [8,13,14,15].

Most cases of post-vaccine optic neuritis are unilateral; however, here, we report a rare case of bilateral optic neuritis after ChAdOx1-S vaccination.

## 2. Case Presentation

A 49-year-old Asian female visited our emergency department due to acute vision loss within 2 days. She had no prior medical history and denied any recent trauma. She had bilateral, extraocular pain while rotating her eyes. Examination revealed that the best-corrected visual acuity (BCVA) in the right eye was 20/30 and in the left eye was 20/200. Intraocular pressure OU was normal, and the anterior chamber was unremarkable. The swinging light test showed a positive relative afferent pupillary defect in the left eye. The Ishihara color vision test revealed 15/15 OU. Fundoscopy showed bilateral disc edema with the left eye more prominent than the right eye (Figure 1A). She had had her first inoculation with the ChAdOx1 COVID-19 vaccine around 2 weeks prior, after which the acute vision loss occurred.

Under suspicion of post-vaccine ON, we performed emergent orbital MRI. T2-weighed MRI revealed significant enhancement of the bilateral optic nerve, more obvious on the left side than on the right side (Figure 1B). No parenchymal lesions were observed. Fluorescein angiography showed bilateral disc leakage in the late phase (Figure 1C). The patient’s blood tests, including serum glucose, complete blood count, and kidney and liver function test returned normal. CRP and ESR were 2.2 mg/L and 26 mm/h, respectively. Immunological assessment covering ANA, anti-DNA, ANCA, anti-Ro, anti-La, and antiphospholipid antibodies returned negative, as did the serum anti-aquaporin-4 (AQP4) antibody test. Unfortunately, a serum anti-myelin-oligodendrocyte-glycoprotein (MOG) antibody test was not commercially available. The PCR test for SARS-CoV-2 based on nasopharyngeal swab was negative. The patient refused lumbar puncture at that time, and therefore, no CSF-related data are available.

The patient was given high-dosage IV steroids (methylprednisolone 1 g/day for 3 days) with subsequent improvement in visual acuity from day 2, followed by oral prednisolone (1 mg/kg), tapering over 2 weeks. Seven weeks later, the patient’s best-corrected visual acuity in both eyes was 20/20 and the papillitis was resolved. The timeline of the clinical course is shown in Figure 2.

## 3. Discussion and Conclusions

We report a case of bilateral ON, occurring two weeks after the patient’s first dose of the ChAdOx1-S vaccine. The case involved the sudden onset of vision decline in both eyes with the typical presentation of ON: extraocular pain during eye movement. However, color vision remained normal; in this instance, the case differs from typical idiopathic ON due to intact color vision. This patient had no medical history, which confirmed the temporal relationship between the vaccine and ON. Most cases of vaccine-related ON occur 2 to 3 weeks after vaccination. Some patients’ vision decline is subtle, while others’ is dramatic, depending on how much of the optic nerve is involved. ON is a known but rare adverse side effect of vaccination, the exact mechanism of which is not clear. Possibly, the vaccine’s activation of the host immune system leads to T-cell activation, which then damages the optic nerve’s myelin sheath [16].

According to previous literature, the incidence of optic neuritis is 2.6–3.9 per 10,000 adult persons [17] and 1.79–2.46 per 100,000 persons in the pediatric population [18]. The incidence of vaccine-related optic neuritis has not been reported because neuro-ophthalmic presentations related to vaccination are extremely rare [19]. In the case of a relatively new vaccine, such as the COVID-19 vaccine, any reports of adverse effects after vaccination should be collected to comprehensively assess side effects. Case reports and case series related to post-vaccine optic neuritis have been published recently because more physicians have become aware of this possible side effect [8,9,14,20,21,22,23,24,25]. However, since the causative relationship between vaccination and idiopathic optic neuritis is difficult to delineate, in the case of our patient, we were clinically left with only the temporal association since she also had no medical history, and the most common period for vaccine-related immune reactions is two to three weeks after inoculation [26].

In Taiwan, five vaccines have been issued: ChAdOx1-S (AstraZeneca, AZ), BNT162b2 (BNT), mRNA-1273 (Moderna), Nuvaxoid (Novavax), and MVC-COV1901(Medigen). Since the first outbreak in May 2021, a large population has been vaccinated with the ChAdOx1-S (AstraZeneca) vaccine because this was the only vaccine our government could offer at that time. As more vaccines were used and lower-aged groups were vaccinated, more information on post-vaccine adverse effects was reported.

In a report dated 19 June 2021, the Taiwan Central Epidemic Command Center (CECC) revealed that a total of 1,374,956 AstraZeneca shots had been administered in Taiwan up to 18 June, of which 269,056 were given to the elderly, aged 75 years and older. To date, the Vaccine Adverse Event Reporting System (VAERS) has registered 49 reports of death following ChAdOx1-S vaccination. At the time of writing this manuscript, the detailed VAERS report could be accessed through the website of the Taiwan Center for Disease Control (https://www.cdc.gov.tw/en/Disease/SubIndex/ (accessed on 28 September 2022)) (Table 1).

Recently, two cases of acute zonal occult outer retinopathy (AZOOR) and bilateral arteritic anterior ischemic optic neuropathy (AAION) after COVID-19 m-RNA vaccination were reported [27]. Four cases of post-vaccine ON were reported in a case series, three of which were unilateral ON and the remaining one bilateral ON. However, these four cases regained their vision after prompt treatment [14]. ON may result from autoimmune, infectious, or inflammatory disorders. ON is one of the most commonly seen neurological adverse effects after vaccines, including measles, influenza, hepatitis A/B, pneumococcal vaccine, human papilloma virus, and rabies vaccines [3,4,5,6,28,29,30,31,32,33,34,35,36,37,38,39]. The mechanisms for these inflammatory/autoimmune events are mostly unknown, but some pathways are presumed. First, molecular mimicking, based on COVID-19-related autoimmune reactions to viral proteins resulting from mRNA translation, which induce immune interactions with human cells. Second, some vaccine adjuvants may activate the NLRP3 inflammasome, leading to inflammation and immunity [40].

Our case had bilateral ON temporally associated with ChAdOx1-S (AstraZeneca) vaccination. The patient responded well to corticosteroid therapy. Her ON and visual acuity recovered soon after the treatment. However, given the demyelinating nature of ON, other demyelination diseases, such as NMOSD, MOG-ON, or MS, might occur in the future. Patients with post-vaccine ON should be educated to become aware of any dangerous signs/symptoms of demyelination disorders for the patient still has the possibility to develop any of the above kinds of demyelination disorders [41,42].

## Figures and Tables

**Figure 1 vaccines-10-01889-f001:**
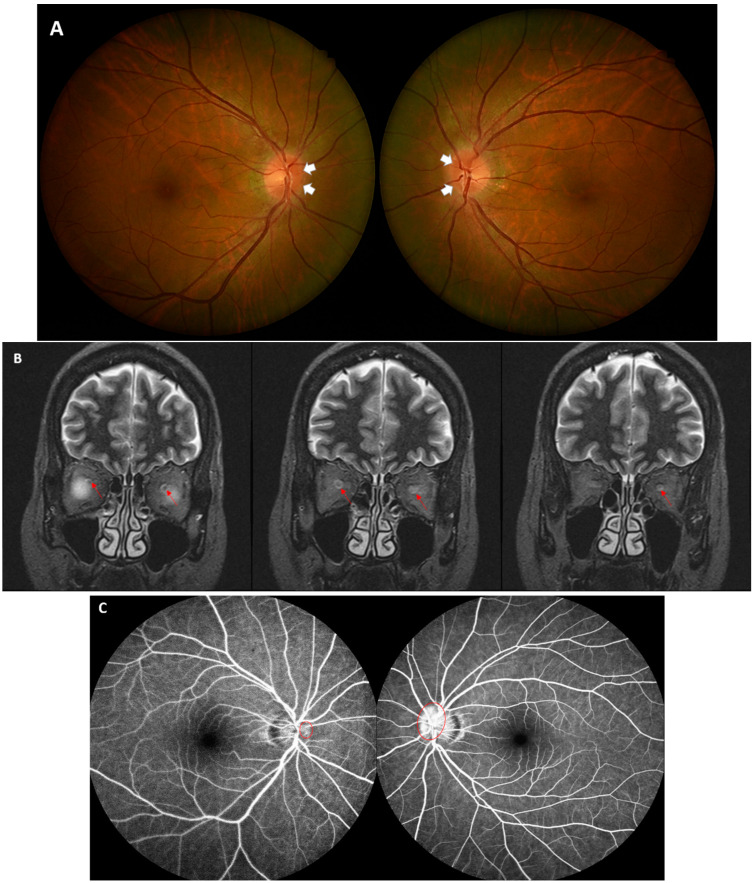
(**A**) Fundus color photography revealed bilateral disc edema (white arrows), with the left side more prominent. (**B**) Coronal T2-weighted images of the orbits showing T2 magnetic resonance image in the bilateral optic nerves, more profound on the left side (red arrows). (**C**) Fluorescein angiography showed late phase dye leakage (red circles), confirming disc edema without vessel abnormalities.

**Figure 2 vaccines-10-01889-f002:**
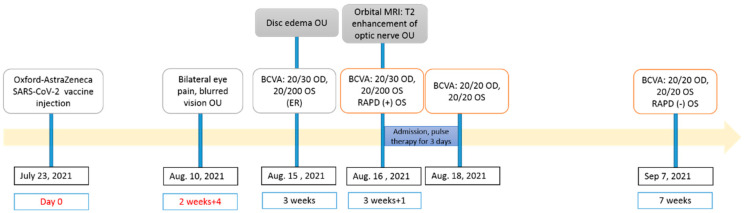
The timeline of our patient’s clinical presentation.

**Table 1 vaccines-10-01889-t001:** Post-COVID 19 Vaccine Adverse Effect Report (AE = adverse effect).

Vaccine		Severe AE	Non-Severe AE	Total	Death	Suspected Severe Allergic Reaction	Other Suspected AE
Total	110/3/22~111/8/18	20,355	9957	10,398	1544	45	8809
AstraZeneca	110/3/22~111/8/18	8526	4172	4354	850	25	3479
Moderna	110/6/8~111/8/18	5359	2311	3048	508	11	2529
Medigen	110/8/23~111/8/18	814	415	399	57	6	336
BioNTech	110/9/22~111/8/18	5643	3054	2589	127	3	2459
Novavax	111/7/8~111/8/1	13	5	8	2	0	6

## Data Availability

Data are available upon request.

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
