# Peer review of "Bilateral Optic Neuritis after COVID-19 Vaccination: A Case Report"

_vaccines, 2022, doi:10.3390/vaccines10111889_

Round 1
Reviewer 1 Report
A temporal association does not necessarily mean a causative association. As millions have received the vaccine during the respective period it may be just a coincidental occurrence. What is the background incidence of bilateral ON at the same time in the population? This case report shows no convincing association of ON and the COVID-19 vaccination. The case presentation should include the visual fields. VEP would be useful as well. Ishiara is not a good test for acquired color vision deficits. Panel-D-15 or a similar test may be better.
Author Response
Thank you for your comment. Though most patients with optic neuritis are idiopathic, we still believe this case was probably related to the preceding vaccine. Vaccine-related optic neuritis is a very rare event, but when it occurs it arises within 1 to 3 weeks after vaccination. Previous literature puts the incidence of pediatric optic neuritis in Taiwan at around 1.79-2.46 per 100,000 persons [1], and the incidence of adult optic neuritis at 2.6-3.9 per 10,000 persons [2]. Causal association between the vaccine and optic neuritis can only be based on temporal association, because, as the reviewer points out, idiopathic optic neuritis occurs on an ongoing basis and this might be a coincidence. However, according to a review [3], our case was an atypical case of optic neuritis, and the patient had no prior medical history. Clinically, we attributed the ON to COVID-19 vaccination because most vaccine-related side effects have been reported with a similar temporal association [4-14]. For the newly launched vaccines of the COVID-19 pandemic era, we must collect all possible data and gather information for post-marketing surveillance.
We regret that we couldn’t obtain a timely visual field report and VEP for the patient because the examinations were scheduled very late due to the impact of COVID-19 infection and the shortage of our staff at that time. The patient received pulse therapy from 2021/08/16 to 2021/08/18, while visual field examination was done on 2021/08/24 and VEP was done on 2021/08/31. We therefore don’t show the results. Panel-D-15 or Farnsworth Munsell 100 hue test are truly better color vision tests, however, in our busy clinical setting we elected to use the Ishihara test to screen for any acquired color vision deficit.
Please see the attachment.
References
- Lin WS, Wang HP, Chen HM, Lin JW, Lee WT: Epidemiology of pediatric multiple sclerosis, neuromyelitis optica, and optic neuritis in Taiwan. Journal of neurology 2020, 267(4):925-932.
- Woung LC, Lin CH, Tsai CY, Tsai MT, Jou JR, Chou P: Optic neuritis among National Health Insurance enrollees in Taiwan, 2000-2004. Neuroepidemiology 2007, 29(3-4):250-254.
- Abel A, McClelland C, Lee MS: Critical review: Typical and atypical optic neuritis. Survey of ophthalmology 2019, 64(6):770-779.
- Liu Z, Zhang L, Yang Y, Meng R, Fang T, Dong Y, Li N, Xu G, Zhan S: Active Surveillance of Adverse Events Following Human Papillomavirus Vaccination: Feasibility Pilot Study Based on the Regional Health Care Information Platform in the City of Ningbo, China. J Med Internet Res 2020, 22(6):e17446.
- Jun B, Fraunfelder FW: Atypical Optic Neuritis After Inactivated Influenza Vaccination. Neuro-ophthalmology (Aeolus Press) 2018, 42(2):105-108.
- Netravathi M, Dhamija K, Gupta M, Tamborska A, Nalini A, Holla VV, Nitish LK, Menon D, Pal PK, Seena V et al: COVID-19 vaccine associated demyelination & its association with MOG antibody. Multiple sclerosis and related disorders 2022, 60:103739.
- García-Estrada C, Gómez-Figueroa E, Alban L, Arias-Cárdenas A: Optic neuritis after COVID-19 vaccine application. Clin Exp Neuroimmunol 2021.
- Hull TP, Bates JH: Optic neuritis after influenza vaccination. American journal of ophthalmology 1997, 124(5):703-704.
- Agarwal A, Garg D, Goyal V, Pandit AK, Srivastava AK, Srivastava MP: Optic neuritis following anti-rabies vaccine. Tropical doctor 2020, 50(1):85-86.
- Roy M, Chandra A, Roy S, Shrotriya C: Optic neuritis following COVID-19 vaccination: Coincidence or side-effect? - A case series. Indian journal of ophthalmology 2022, 70(2):679-683.
- O'Brien P, Wong RW: Optic neuritis following diphtheria, tetanus, pertussis, and inactivated poliovirus combined vaccination: a case report. Journal of medical case reports 2018, 12(1):356.
- Han SB, Hwang JM, Kim JS, Yang HK: Optic neuritis following Varicella zoster vaccination: report of two cases. Vaccine 2014, 32(39):4881-4884.
- Monge Galindo L, Martínez de Morentín AL, Pueyo Royo V, García Iñiguez JP, Sánchez Marco S, López-Pisón J, Peña-Segura JL: Optic neuritis in paediatric patients: Experience over 27 years and a management protocol. Neurologia (Engl Ed) 2021, 36(4):253-261.
- Elnahry AG, Asal ZB, Shaikh N, Dennett K, Abd Elmohsen MN, Elnahry GA, Shehab A, Vytopil M, Ghaffari L, Athappilly GK et al: Optic neuropathy after COVID-19 vaccination: a report of two cases. Int J Neurosci 2021:1-7.

Reviewer 2 Report
Many thanks to the authors who worked hard on this article. I would like to thank the MDPI editors who invited me to review this manuscript which aimed to evaluate the temporal relationship between vaccination and the occurrence of ON and report a case of bilateral ON after the administration of the ChAdOx1-S nCoV- 14 19 SARS-CoV-2 vaccine.
In my opinion, the section conclusion should be added
Author Response
Thank you for your suggestion. We have added further information in the conclusion section, highlighted in red at lines 113-124 on page 4.
Please see the attachment.

Reviewer 3 Report
Thanks to the authors to present a case of bilateral ON after Astrozeneca covid 19 vaccine.
Some scientific concers have been raised:Missing data, Missinfg new and update references,No available CSF,method of antiaquqpurine test,Missing Brain MRI slices,misssing follow and no explanation of recovery and its degree...
Suggestion:
Provide the data , missing ones and follow the patientThen reconsider to be submitted
Author Response
Thank you for your advice. We have updated our references. Regarding the CSF data, the patient refused lumbar puncture at the time, so we have no related data. Also, we have added some information about AQP4 and the reason for the lack of CSF data in the original manuscript at line 83-87, on page 3. We have also added more brain MRI slices as per your suggestion. The patient failed to hold the appointment for further follow-up.
Please see the attachment.

Round 2
Reviewer 1 Report
ok
Author Response
Thank you for your recognition.
Reviewer 3 Report
Thanks to clarify the issuese.
Only pls write in the subtitle of imaging, T2 MRI not T2 gadilinum enhancing.
Also in the discussion emphasis that the probability of MOGAD ,O.N and aquaporin ab negative O.N. could not be exactly ruled out.
Author Response
1. Only pls write in the subtitle of imaging, T2 MRI not T2 gadilinum enhancing.
Reply: Thank you for the comment. We already revised this at line 89 in page 3 (in green color).
2. Also in the discussion emphasis that the probability of MOGAD ,O.N and aquaporin ab negative O.N. could not be exactly ruled out.
Reply: Thank you for your advice. We did emphasize this point in the end of our discussion at line 155-159 in page 5 (in green color).
